# Recent Progress in Systemic Therapy for Advanced Hepatocellular Carcinoma

**DOI:** 10.3390/ijms25021259

**Published:** 2024-01-19

**Authors:** Narayanan Sadagopan, Aiwu Ruth He

**Affiliations:** MedStar Georgetown Lombardi Comprehensive Cancer Center, Washington, DC 20007, USA; aiwu.r.he@gunet.georgetown.edu

**Keywords:** hepatocellular carcinoma, immunotherapy, VEGF, tyrosine kinase inhibitors, systemic treatment, advanced HCC, CAR T-cells

## Abstract

Patients with advanced hepatocellular carcinoma (HCC) have several systemic treatment options. There are many known risk factors for HCC, and although some, such as hepatitis C, are now treatable, others are not. For example, metabolic dysfunction-related chronic liver disease is increasing in incidence and has no specific treatment. Underlying liver disease, drug resistance, and an increasing number of treatment options without specific biomarkers are all challenges in selecting the best treatment for each patient. Conventional chemotherapy is almost never used for advanced-stage disease, which instead is treated with immunotherapy, tyrosine kinase inhibitors, and VEGF inhibitors. Immune checkpoint inhibitors targeting various receptors have been or are currently undergoing clinical evaluation. Ongoing trials with three-drug regimens may be the future of advanced-stage HCC treatment. Other immune-modulatory approaches of chimeric antigen receptor-modified T cells, bispecific antibodies, cytokine-induced killer cells, natural killer cells, and vaccines are in early-stage clinical trials. Targeted therapies remain limited for HCC but represent an area of potential growth. As we shift away from first-line sorafenib for advanced HCC, clinical trial control arms should comprise a standard treatment other than sorafenib, one that is a better comparator for advancing therapies.

## 1. Introduction

Hepatocellular carcinoma (HCC) is not a conventional solid tumor malignancy. Traditional systemic chemotherapy drugs are not very effective, possibly due to higher expression of drug resistance genes and due to baseline liver dysfunction [1]. Unlike most malignancies, HCC can be diagnosed without a tissue biopsy and just with imaging. A four-phase CT scan of the abdomen with IV contrast or a liver MRI with contrast can diagnose HCC in patients with cirrhosis or chronic hepatitis B [2]. A four-phase CT scan can have a sensitivity as high as 90% for HCC greater than 2 cm in diameter [3]. While this can initially benefit the patient by avoiding an invasive diagnostic procedure, modern cancer work-up often involves sending tissue samples for next-generation sequencing to look for targetable mutations or markers of responsiveness to treatment. Limited tissue sampling may have hampered some advancements in HCC therapy.

Treatment, as always in cancer, begins with staging, although in this respect HCC is once again unique. The Barcelona Clinic Liver Cancer (BCLC) staging system is the most used approach. It incorporates Eastern Cooperative Oncology Group (ECOG) performance status; Child–Pugh class, which is an estimate of mortality in cirrhosis patients; size and number of tumors; vascular involvement; and extrahepatic spreading of the tumor [4]. Barcelona stage B (intermediate stage) or stage C (advanced stage) patients may be eligible for systemic treatment, which is the focus of this review. Due to the ineffectiveness of chemotherapy, most of the current approved systemic treatments for HCC are tyrosine kinase inhibitors (TKI), vascular endothelial growth factor (VEGF) inhibitors, and immunotherapy agents [5]. Resection, liver transplant, ablation, transarterial chemoembolization (TACE), transarterial radioembolization (TARE), and hepatic artery infusion (HAI) are all localized HCC treatment options that will not be discussed in this review.

The main risk factors for HCC are chronic liver disease via hepatitis B infection, hepatitis C infection, metabolic dysfunction-associated steatotic liver disease (MASLD), and chronic alcoholism. All these factors play a role in inflammation-related carcinogenesis, which can contribute to >90% of HCC cases. Chronic inflammation leads to hepatic cell death and concurrent regeneration with the pro-inflammatory microenvironment leading to dysplastic hepatocyte proliferation and eventually malignancy [6]. The significance of any one of these factors varies considerably across the globe. Worldwide, hepatitis B is the most common cause of HCC-related death, but alcohol- and hepatitis C-related HCC are more common in the United States [7]. With about 900,000 new cases of HCC recorded globally in 2020—a number that is expected to keep on growing—it is crucial to continue improving systemic therapies for HCC [8]. There is no recommended routine screening method for HCC in the general population. Patients with cirrhosis and high-risk hepatitis B are advised to undergo surveillance ultrasound every 6 months [9].

Catching HCC in its early stages would be optimal, but having good treatment options for later-stage disease is just as important. How to best select treatment options for each specific advanced HCC patient with a lack of biomarker data and an increasing amount of treatment options remains one of the biggest challenges. One aspect is many patients often have underlying liver dysfunction, which can be a significant issue depending on how a drug is metabolized. Another aspect is the presence of multidrug resistance protein family transporters on HCC cells, which is among the contributing factors to drug resistance [10]. Lastly, the number of treatment options has increased dramatically over the past 5 years (Figure 1). While this is generally a good thing, the lack of clinical trial data comparing agents aside from sorafenib makes treatment selection more difficult. This article will discuss currently approved therapies and novel treatment approaches that are presently under clinical study, and which have the potential to reshape the HCC treatment landscape.

## 2. Evolution of First-Line Treatment

In 2007, the TKI sorafenib was the first systemic therapy to show benefits for advanced HCC with a median overall survival (OS) of 10.7 months compared to 7.9 months for placebo [11]. In 2018, the TKI lenvatinib was shown to be non-inferior to sorafenib in the REFLECT trial, which was an open-label phase III trial of 954 patients. The median OS of patients treated with lenvatinib was 13.6 months vs. 12.3 months for sorafenib [12].

The landmark IMbrave150 trial published in 2020 switched the initial systemic treatment approach for advanced HCC away from TKIs towards immunotherapy. Atezolizumab plus bevacizumab provided a significantly better median OS than sorafenib (19.2 months vs. 13.4 months) and significantly improved progression-free survival (PFS: 6.9 months vs. 4.3 months). Treatment-related grade 3 or 4 adverse events were similar between groups. No patients had grade 3 or 4 pneumonitis or myocarditis, which are among the more significant immune-related adverse events in patients receiving immunotherapy [13,14]. These results established atezolizumab plus bevacizumab as one of the preferred first-line regimens at this time. The other preferred first-line regimen is the immunotherapy doublet tremelimumab plus durvalumab, which came out of the HIMALAYA trial published in 2022. Of note, patients in the tremelimumab plus durvalumab group only received one total dose of tremelimumab each. Tremelimumab plus durvalumab had a median OS of 16.4 months compared to 13.8 months for sorafenib (*p* = 0.0035) [15]. It is interesting to see the median OS of sorafenib increasing in more recent trials, pointing to a generalized improvement in cancer care (Figure 2).

Durvalumab monotherapy is another option for HCC which was found to be non-inferior to sorafenib (median OS 16.6 months, survival HR 0.86, and 95% CI 0.73–1.03) in the HIMALAYA trial. The HIMALAYA trial was designed to test multiple hypotheses with a durvalumab plus tremelimumab group, durvalumab group, and a sorafenib group. The median PFS was not significantly different between all 3 groups. OS at 36 months was 30.7%, 24.7%, and 20.2% for tremelimumab plus durvalumab, durvalumab, and sorafenib, respectively. Durvalumab did have lower grade 3 or 4 treatment-emergent adverse effects at 37.1% compared to 50.5% for tremelimumab plus durvalumab and 52.4% for sorafenib. As commonly seen with immunotherapy doublets, tremelimumab plus durvalumab had more grade 3 or 4 immune-mediated adverse events at 12.6% compared to 6.2% for durvalumab alone [15].

First-line pembrolizumab had promising results in the phase II KEYNOTE-224 trial cohort 2 with a median OS of 17 months, objective response rate (ORR) of 16%, and median duration of response of 16 months [16]. However, the LEAP-002 trial compared pembrolizumab plus lenvatinib to lenvatinib alone in the first-line setting and did not show significant OS or PFS benefits from the addition of pembrolizumab. Despite a prolonged median OS of 21.2 months for pembrolizumab plus lenvatinib, lenvatinib monotherapy had a median OS of 19 months, which was much longer than the OS of 13.6 months in the REFLECT trial, resulting in the LEAP-002 study not meeting its primary OS endpoint with an HR 0.84. Median PFS was similar at 8.2 months for the combination and 8.1 months for lenvatinib alone [17]. Nivolumab, given as a single agent, showed a durable response in a subset of patients in early-phase clinical trials. When nivolumab was tested in the randomized phase III study CheckMate 459, it did not provide a superior survival benefit in comparison to sorafenib [18]. Nivolumab did show a disease control rate of 55% in a phase I/II trial in Child–Pugh class B patients, with a favorable safety profile indicating its benefit in patients with liver dysfunction [19].

## 3. Second-Line and Beyond

There are an increasing number of approved first-line treatment options, while all second-line treatment options were tested in patients who received first-line sorafenib treatment. There is no guideline on how to prioritize and sequence all the treatment options. In practice, many of the first-line options are used in the later-line setting, assuming the patient has not received those agents yet. This pushes the approved second-line treatment into later lines of therapy if patients are still in good shape to receive these treatments. The approved second-line and beyond options are included in Table 1. Regorafenib and cabozantinib are both mixed TKIs, and their common side effects are hypertension, hand-foot syndrome, fatigue, and diarrhea [20,21]. Ramucirumab, a monoclonal antibody to VEGFR2, was explicitly approved for patients with alpha-fetoprotein (AFP) greater than 400 ng/mL [22]. KEYNOTE-240 and KEYNOTE-394 both looked at pembrolizumab in the second-line setting. However, KEYNOTE-394 involved only patients in Asia, while KEYNOTE-240 studied a more diverse population [23,24]. The multi-arm phase I/II study CheckMate 040 compared different dosing regimens of nivolumab and ipilimumab without a control arm. The 22.8-month median OS was from the group who received nivolumab 1 mg/kg plus ipilimumab 3 mg/kg every 3 weeks for 4 cycles. This regimen was followed by nivolumab 240 mg every 2 weeks. That group also had an ORR of 32% [25]. It should be noted that all the trials mentioned in Table 1 are based on patients having progression on sorafenib, which is currently not the first-line standard of care.

## 4. Completed Phase III Trials

With immunotherapy and TKIs independently showing success in HCC, the combination of the 2 have been evaluated in many phase III HCC clinical trials. VEGF, in addition to its potent role in angiogenesis, is also an immunosuppressive factor, so blocking this factor with a TKI could potentiate the effect of immunotherapy [26]. Camrelizumab is an PD-1 antibody, while rivoceranib (also known as apatinib) is a VEGFR-2 targeting TKI. That combination was compared to sorafenib in the first-line setting in a trial involving 543 patients. Median PFS was 5.6 months for the 2 drugs vs. 3.7 months for sorafenib (HR 0.52, 95% CI 0.41–0.65) and median OS was 22.1 months for the 2 agents vs. 15.2 months for sorafenib (HR 0.62 and 95% CI 0.49–0.8). The side effect profile was as expected for the camrelizumab plus rivoceranib group, with the most common grade 3 or 4 adverse effects being hypertension (38%), increase in aspartate aminotransferase (17%), increase in alanine aminotransferase (13%), and hand-foot syndrome (12%). The trial population was about 83% Asian, about 75% HCC due to hepatitis B, and only Child–Pugh class A, which limits its generalizability somewhat [27]. Nonetheless, camrelizumab plus rivoceranib may soon be another first-line HCC treatment option.

The COSMIC-312 trial looked at a combination of 2 approved agents, atezolizumab and cabozantinib. The trial was conducted in the first-line setting with patients who were Child–Pugh class A. The 3 groups consisted of cabozantinib plus atezolizumab, sorafenib, and cabozantinib. The study showed a median PFS of 6.8 months for the combination group compared to 4.2 months for sorafenib (HR 0.63 and 99% CI 0.44–0.91). The initial reported median OS was not different at 15.4 months for the combination compared to 15.5 months for sorafenib. It was noted that the patients in the sorafenib arm received more subsequent lines of treatment versus the combined group, which likely played a role in the median OS result of the control arm. Final longer-term OS data is pending for this trial. Hypertension, increased AST/ALT, and hand-foot syndrome were again the most common serious adverse events [28].

Sintilimab, a PD-1 inhibitor, with IBI305, a bevacizumab biosimilar, was approved in China as a first-line HCC treatment after favorable results from the ORIENT-32 trial. In this trial, a combination of sintilimab plus IBI305 was compared to sorafenib in a trial population of 595 patients. The 2-drug regimen significantly improved the median PFS (4.6 months vs. 2.8 months) and OS (median not reached vs. 10.4 months). This trial took place in 50 clinical sites in China, and 94% of the patients had HCC due to hepatitis B [29]. Fundamentally, this regimen is more or less the same as atezolizumab plus bevacizumab, with the only difference being the target on the PD-1-PD-L1 axis. Atezolizumab targets PD-L1.

Tislelizumab, an anti-PD-1 monoclonal antibody, was compared to sorafenib in the phase III RATIONALE-301 trial in the first-line setting [30]. Tislelizumab was designed to minimize binding to Fc gamma receptors on macrophages to prevent macrophage-mediated destruction of T cells [31]. Tislelizumab was found to have non-inferior median OS compared to sorafenib (15.9 months vs. 14.1 months, HR 0.85, and 95% CI 0.71–1.02). Tislelizumab had a higher ORR of 14.3% compared to 5.4%. Tislelizumab had lower rates of grade 3 and 4 adverse events as well as a lower rate of adverse events leading to drug discontinuation. Hepatitis and hypothyroidism were the most common immune-mediated side effects, both at 5.3% [30].

ADI-PEG 20 was a different HCC treatment approach in the second-line setting. ADI-PEG 20 is a cloned arginine-degrading enzyme. HCC typically lacks argininosuccinate synthetase, which is required to metabolize citrulline to arginine. With HCC cells being unable to produce arginine and with external arginine being broken down by ADI-PEG 20, the tumor cells would not have access to arginine. In a study of 635 patients, ADI-PEG 20 was compared to placebo; there was no significant difference in median OS (7.8 months vs. 7.4 months) or PFS (2.6 months vs. 2.6 months) [32]. Although ADI-PEG 20 did not show any survival benefit over placebo, its tolerable safety profile warranted further investigation in HCC patients with high arginine levels [33].

## 5. Future Directions

Immune checkpoint inhibition is far more complicated than just PD-1 and CTLA-4 (Figure 3). Many more co-stimulatory and co-inhibitor receptors can be targeted [34]. The liver is a very immunologically tolerant organ to limit hypersensitivity to the many antigens it is exposed to in the portal venous system, but this immunosuppressive microenvironment can prevent identification and destruction of HCC, which is why immunotherapy treatments are used [35]. Other aspects of the immune system, such as natural killer (NK) cells, can also be harnessed in the fight against tumor cells. In addition, finding new ways to help the immune system target cancer cells offers another viable solution. The focus is then on defining the right combination of these therapies to yield the most benefit with the least toxicity. The currently approved HCC treatments are either single drugs or two-drug combinations. Other solid tumor malignancies are sometimes treated with three to four agents, which could be a viable strategy in the treatment of HCC as well. Of course, when adding further chemotherapy or immunotherapy, the risk of increased side effects must be weighed against response.

### 5.1. Three-Drug Regimens

Nivolumab plus cabozantinib with or without ipilimumab was one triplet examined in a phase I/II trial. Patients could be untreated or could have received sorafenib in the past. In a group of 71 patients, the ORR was 17% for the doublet and 29% for the triplet. The median PFS and OS of 5.1 months and 20.2 months, respectively, for the doublet compared to 4.3 months and 22.1 months, respectively, for the triplet. There were more side effects in the three-drug group, with 74% of patients having grade three or four adverse effects compared to 50% for the two-drug group [36]. However, the side effects were generally manageable, making the three-drug regimen a potential option for later-stage studies. There is an ongoing phase II trial of nivolumab, cabozantinib, and ipilimumab followed by TACE in HCC patients who are not candidates for curative intent treatment [37]. A similar approach using co-formulated pembrolizumab/quavonlimab with lenvatinib, where quavonlimab is a CTLA-4 inhibitor, is in a phase II trial [38].

Adding another drug to the proven atezolizumab plus bevacizumab treatment is also a reasonable option. The MORPHEUS-liver study was a multi-cohort phase Ib. The investigators tried multiple doublet and triplet combinations using an atezolizumab plus bevacizumab framework, including one cohort with atezolizumab, bevacizumab, and tiragolumab [39]. TIGIT is an inhibitory receptor that can be upregulated by tumor cells and other cells in the tumor microenvironment [40]. Tiragolumab is a TIGIT inhibitor. The concept is like combining PD-1 inhibition with CTLA-4 inhibition. The triplet had an ORR of 42.5% compared to 11.1% in atezolizumab plus bevacizumab. Median PFS was also improved with the triplet at 11.1 months vs. 4.2 months. All results were independent of PD-L1 status, with similar safety profiles in both groups [39]. Building on these results, a phase III trial for atezolizumab plus bevacizumab plus tiragolumab in the first-line setting is currently underway [41]. A phase II trial similar in concept using atezolizumab plus bevacizumab with SRF388, which is a fully human IgG1 blocking immunosuppressive cytokine interleukin 27 (IL-27), is also underway [42]. In addition, the triplet of atezolizumab, bevacizumab, and ipilimumab is being evaluated in the first-line setting in a phase II/III trial [43].

Dual immune checkpoint blockade with the addition of bevacizumab can be approached in multiple ways. Relatlimab is a LAG-3 blocking antibody. Blocking LAG-3 helps upregulate T-cell function [44]. The three-drug regimen of nivolumab, relatlimab, and bevacizumab is being evaluated in a phase I/II trial [45]. Another combination undergoing phase I study is IBI310 (a CTLA-4 inhibitor), sintilimab, and bevacizumab [46]. A phase II trial of the TIGIT inhibitor, ociperlimab, the bevacizumab biosimilar, BAT1706, and tislelizumab is also ongoing [47].

Despite the lack of efficacy of chemotherapy for HCC, there was one trial that evaluated camrelizumab with 5-fluorouracil, leucovorin, and oxaliplatin (FOLFOX) for treatment-naïve advanced HCC patients. It was a phase Ib/II with a total of 34 patients. Grade three or higher adverse events occurred in 85.3% of the patients, with decreased neutrophil count being the most common. The ORR was 29.4%, the disease control rate was 79.4%, the median PFS was 7.4 months, and the median OS was 11.7 months. These results point to some degree of antitumor activity [48]. They also pose the question of whether conventional chemotherapy can work for HCC if it is augmented by immunotherapy. This construct of chemotherapy plus immunotherapy is quite common in the treatment of many other cancer types. A phase III trial using camrelizumab plus FOLFOX is being conducted [49].

### 5.2. Two-Drug Regimens

Various two-agent combinations selected from the immunotherapy, TKI, and VEGF inhibitor pool are being tried in first- and second-line settings (Table 2). Nofazinlimab is a monoclonal antibody targeting PD-1 [50]. AK105, also known as penpulimab, is an antibody targeting PD-1 and anlotinib is a TKI [51].

Another two-drug regimen undergoing phase II trial is MTL-CEBPA with sorafenib, tested against sorafenib alone in the second-line setting [58]. Transcription factor CCAAT/enhancer-binding protein alpha (C/EBP-α) is a master regulator of liver homeostasis, myeloid function, and multiple oncogenic processes including cell cycle control, proliferation, and angiogenesis. MTL-CEBPA is a small activating RNA that upregulates C/EBP-α, resulting in inhibition of tumor growth. In a completed phase I study of MTL-CEBPA in HCC, 9 out of 34 patients had grade three treated-related adverse events, and one patient had a partial response lasting longer than 2 years [59].

Pexa Vec with sorafenib is a slightly different version of immunotherapy plus TKI. Pexa Vec is a vaccinia virus-based oncolytic immunotherapy that is supposed to replicate in and destroy tumor cells preferentially. In a phase II trial for HCC patients who failed sorafenib, Pexa Vec did not improve overall survival compared to best supportive care but generally had a tolerable safety profile with 8% of the patients having grade three fever and 8% having grade three hypotension [60]. A phase III trial in the first-line setting is currently comparing Pexa Vec with sorafenib to sorafenib alone [61].

### 5.3. Chimeric Antigen Receptor-Modified T Cells (CAR T-Cells)

CAR T-cells are genetically modified T cells with fusion proteins that target a molecule on the tumor cell surface. CAR T-cells were initially established as a treatment modality for acute lymphoblastic leukemia (ALL) and have gone on to be incorporated into the treatment of multiple hematologic malignancies [62]. The use of CAR T-cells in solid tumors is very rare. Still, they represent an option if viable targets are established.

Glypican-3 (GPC3) is one such target as it is expressed in 75% of HCC but not in normal tissue (Table 3). Success targeting GPC3 with CAR T-cells was seen in vitro, in mouse models, and in patient-derived xenograft models [63,64]. A completed phase I trial in 13 patients using GPC3-targeted CAR T-cells resulted in two patients having partial responses and one having stable disease [65]. AFP also represents a potential target as early-stage models have shown activity targeting it with CAR T-cells [66]. Other CAR T-cell targets in HCC include melanoma antigen gene family (MAGE), New York esophageal squamous cell carcinoma 1 (NY-ESO-1), epithelial cell adhesion molecule (EPCAM), human telomerase reverse transcriptase (hTERT), viral surface antigens in HCC associated with hepatitis B or C, and NK group 2 member D ligand (NKG2DL) [67,68,69,70,71].

### 5.4. Bispecific Antibodies

Bispecific antibodies like CAR T-cells need a target on the cancer cell. The same group of potential targets mentioned above can again be considered in this case. Bispecific antibodies targeting GPC3 have shown activity in xenograft mice [84]. ERY974 is a bispecific antibody that engages CD3 on T cells and GPC3 on tumor cells that underwent a phase I trial in patients with any GPC3-positive solid tumor type. Of the 29 subjects enrolled, only one patient with esophageal cancer demonstrated a partial response [85].

### 5.5. Cytokine-Induced Killer (CIK) Cells and NK Cells

CIK cells are a mix of T-cells and NK cells which are created by incubation of a patient’s peripheral blood mononuclear cells with IL-2 and an antibody against CD3. CIK cells are more specific for tumor cells than normal cells, preferentially leading to tumor cell death. In the adjuvant setting, a phase III trial of CIK cells versus no adjuvant treatment in 230 HCC patients who underwent curative intent treatment showed a median recurrence-free survival of 44 months following CIK cell treatment compared to 30 months in the control group (HR 0.63 and 95% CI 0.43–0.94). All-cause mortality also favored the CIK cell group (HR = 0.21 and 95% CI 0.06–0.75). The CIK cell group had significantly more adverse events at 62% versus 41% (*p* = 0.002) with pyrexia, fatigue, and upper respiratory tract infections being some of the more common ones. Still, there was no significant difference in serious adverse events [86].

NK cells are part of the innate immune response and are capable of directly killing cancer cells. Cytokines from NK cells contribute to a level of inflammation in the tumor microenvironment. Again, in the adjuvant setting, five patients received ex vivo-expanded allogenic NK cells following hepatic resection for HCC. There were no adverse events. Four patients were alive at 3 years, but two had recurrence at the one-year mark [87]. Due to the small sample size, it is difficult to comment on the efficacy of NK cells from this study. However, NK cells showed a promising safety profile. Some of the ongoing NK cell trials are listed in Table 4, including a few with chimeric antigen receptor NK (CAR-NK) cells.

### 5.6. Targeted Agents

When considering targeted therapy aimed at one driver mutation, HCC has much further to go than, say, lung malignancy. As more tumor sequencing data are generated, potential targets should continue to emerge. Fisogatinib, which is an oral FGFR4 inhibitor, is one agent that has completed a phase I trial. FGFR4 is the receptor of FGF19, which may be a driver mutation in some HCCs. In 81 patients, there was a tolerable safety profile with an ORR of 17% in patients positive for FGF19 expression via immunohistochemistry. ORR was 0% in patients who were negative for FGF19 expression [94]. There are currently more early-stage trials involving fisogatinib.

Namodenoson, an A3 adenosine receptor agonist, represents another targeted therapy option. The A3 adenosine receptor is overexpressed in multiple types of solid tumors, including HCC. This receptor has low levels of expression on normal tissue. Namodenoson binds the cellular A3 adenosine receptor and eventually induces apoptosis. In a phase II trial in the second-line setting, there was no significant OS benefit with namodenoson compared to a placebo. However, in a subgroup analysis of patients with a Child–Pugh score of seven, 12-month OS favored namodenoson at 44% vs. 18% for the placebo (*p* = 0.028) [95]. This study is being followed by a phase III trial looking at namodenoson, specifically in HCC patients with a Child–Pugh score of seven [96].

### 5.7. Vaccines

Vaccines against tumor antigens are yet another approach to help the immune system detect and kill cancer cells. Epitopes from AFP and GPC-3 have been used in vaccines without any noteworthy clinical results [97,98]. Peptide-based vaccines and dendritic cell-based vaccines have undergone trials [99,100]. Newer technology relating to mRNA vaccines is also being evaluated in HCC [101,102]. Further work is still required to find the right immunogenic cancer antigens. However, immune escape mechanisms in malignancies make it unlikely that single-agent vaccines will be a treatment option anytime soon. Vaccines are now being evaluated with other agents, such as immune checkpoint inhibitors [103].

## 6. Drug Resistance

Normal hepatocytes play a large role in drug metabolism giving HCC inherent drug resistance mechanisms [104]. Mechanisms of drug resistance can be quite complex with increased drug efflux, decreased drug uptake, increased drug metabolism, sequestration of drugs, and non-coding RNA expression all playing a role. Permeability-glycoprotein is a drug efflux protein which when overexpressed in HCC confers resistance to drugs such as 5-fluorouracil and epirubicin [105]. SLC46A3 can be downregulated in around 80% of HCC cells resulting in decreased uptake of sorafenib [106]. MicroRNA-122, which is a small non-coding RNA involved in regulation of gene expression, can be decreased in sorafenib resistant HCC leading to downstream activation of pathways targeted by sorafenib [107]. Immunotherapy resistance mechanisms include cytokines creating an immunosuppressive microenvironment, lack of tumor immunogenicity, or dysfunctional antigen presentation by tumor cells [108]. Increased VEGF-D and angiopoietin-2 (ANG-2) have specifically been implicated in resistance to atezolizumab plus bevacizumab [109]. Numerous other targets causing resistance to advanced HCC treatments have been suggested. While trying to generate a targeted treatment to overcome a specific resistance mechanism remains an option, using multiple drugs with differing mechanisms of action together is an often a simpler way to overcome a complex drug resistance problem.

## 7. Discussion

A significant change in evaluating first-line therapies for HCC will soon be required as atezolizumab plus bevacizumab and tremelimumab plus durvalumab have been shown to be significantly better than sorafenib, and one of these doublets should probably become the new control arm. All the above first-line treatment phase III data mentioned in this article were generated in comparison to sorafenib, which is slowly becoming a drug of the past for upfront advanced HCC treatment and is inappropriate as a comparator moving forward.

A median OS of 19–22 months, an ORR of about 30%, and a well-manageable safety profile can be expected from today’s standard first-line therapy [13,15,27]. The data from not yet reported phase III trials will generate more knowledge on subtle differences in efficacy and toxicities of different combinations. It seems unlikely that any two-drug combination selected from the available immunotherapy, TKI, and VEGF inhibitor pool will do significantly better than the current standard. Additional “me too” clinical trials should be discouraged so resources can be used for innovative treatment or combinations. The future of first-line advanced HCC treatment probably lies in finding the right synergy with inhibition of the PD-1/PD-L1 axis. Bevacizumab plus atezolizumab and tremelimumab plus durvalumab are excellent examples of the synergy between the anti-VEGF and anti PD-/PD-L1 axis and the anti-CTLA-4 and anti PD-1/PD-L1 axis, respectively. Using a three-drug combination targeting multiple immune checkpoints may take a leap from the current efficacy benchmark. Safety of the treatment in a patient population that usually has some degree of liver dysfunction is always as important as the efficacy of the treatment. Furthermore, biomarkers for efficacy and safety are urgently needed in a disease with many treatment options but still a dismal 5-year survival rate of 11% [110].

Other immune-modulating approaches of CAR T-cells, bispecific antibodies, CIK cells, and NK cells are still some ways from having more concrete roles in the later-line setting for advanced HCC. Given the history of these types of immune-modulating agents in solid tumors, it is improbable that they will be a part of the first-line treatment arsenal anytime soon. Despite all the focus on using AFP or GPC3 as targets, the lack of progress in these arenas could mean that an entirely different targeting antigen holds the key to new approvals. HCC vaccines have yet to have a significant breakthrough, but that could come from combination treatment with other agents. Lastly, more targeted agent trials for HCC are probably on the horizon as more is learned about the mutational landscape of HCC. There may be opportunities to individualize treatment using associated risk factors such as hepatitis B or C as well.

In conclusion, most of the approved treatments for advanced-stage HCC in any line of therapy fall into the broad categories of immunotherapy, TKI, or anti-VEGF agent. Many multiple-drug immune-modulating approaches are being evaluated in clinical trials to find the best synergy between the different types of treatment. Median OS for patients with first-line treatment has essentially doubled compared to the 10.7 months from the first sorafenib trial. With increasing amounts of next-generation sequencing data from HCC samples, discovery of reliable biomarkers for treatment response becomes more likely, and this can provide much needed clarity as more and more agents become available. The order of agents in the second-line setting and beyond also needs to be better defined. The outcomes of ongoing triplet regimen trials have the potential to alter first-line HCC treatment in a similar way to the IMbrave150 and the HIMALAYA trials.

## Figures and Tables

**Figure 1 ijms-25-01259-f001:**
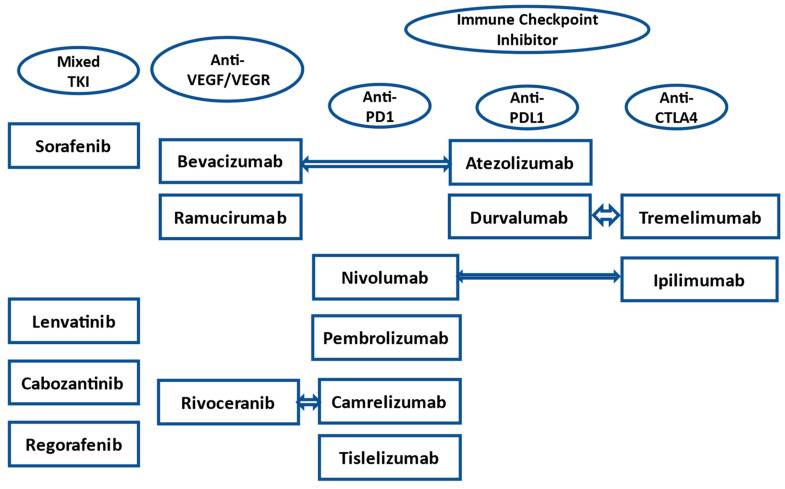
Advanced HCC treatment options from all lines of treatment in the U.S. by drug class. Arrows indicating drugs that are used together. PD-1 = programmed cell death 1. PD-L1 = programmed cell death ligand 1. CTLA-4 = cytotoxic T-lymphocyte associated protein 4.

**Figure 2 ijms-25-01259-f002:**
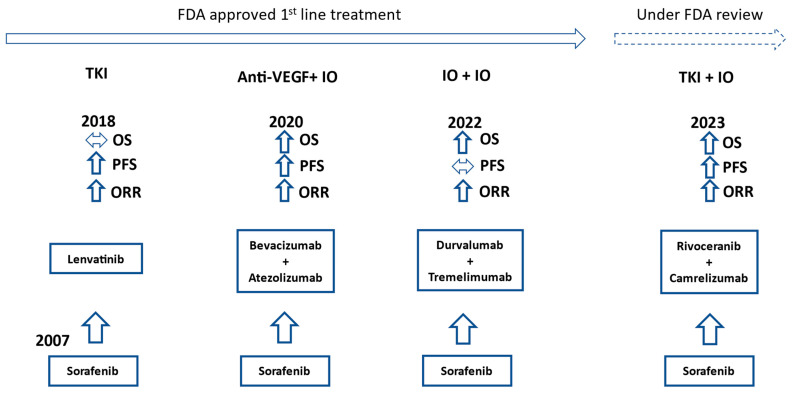
Timeline of FDA approvals for first-line advanced HCC treatment. OS, PFS, and ORR comparisons to sorafenib in their respective trials. IO = immunotherapy.

**Figure 3 ijms-25-01259-f003:**
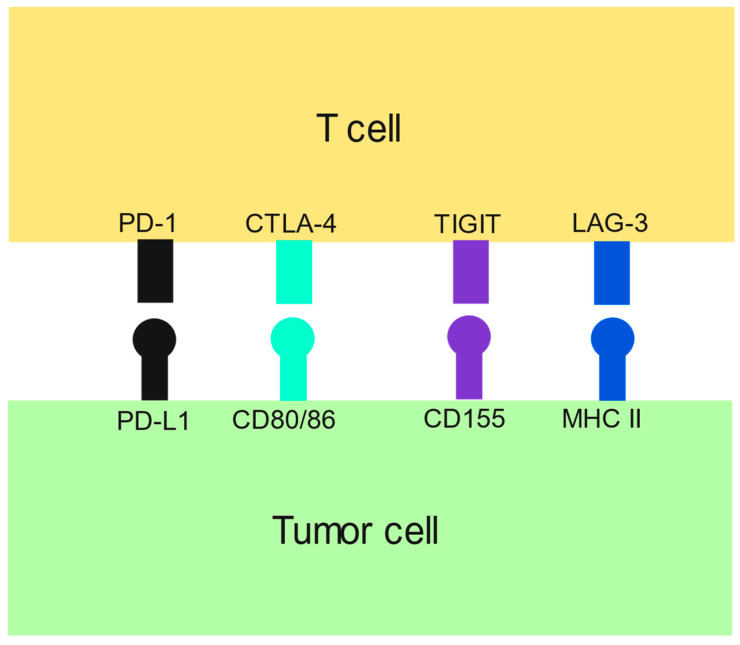
Simplified representation of HCC immunotherapy targets with approvals or under investigation. TIGIT = T-cell immunoreceptor with immunoglobulin and ITIM domain. MHC II = major histocompatibility complex class II. LAG-3 = lymphocyte-activation gene 3. There are numerous other interactions between T cells and tumor cells receptors/ligands not pictured.

**Table 1 ijms-25-01259-t001:** Trial results leading to approval of later-line options for advanced HCC [20,21,22,23,24,25].

Drug (Trial)	Control	Total Number of Patients	Drug Median OS (Months)	Control Median OS (Months)	HR (95% CI)
Regorafenib (RESORCE)	placebo	573	10.6	7.8	0.63 (0.5–0.79)
Ramucirumab (REACH-2)	placebo	292	8.5	7.3	0.71 (0.53–0.95)
Cabozantinib (CELESTIAL)	placebo	707	10.2	8	0.76 (0.63–0.92)
Pembrolizumab (KEYNOTE-240)	placebo	413	13.9	10.6	0.78 (0.61–0.998)
Pembrolizumab (KETNOTE-394)	placebo	453	14.6	13	0.79 (0.63–0.99)
Nivolumab + Ipilimumab (CheckMate 040)	n/a	148	22.8	n/a	n/a

**Table 2 ijms-25-01259-t002:** Two-drug regimens in clinical trials [50,51,52,53,54,55,56,57].

Trial ID	Medications	Comparison	Line of Therapy	Phase
NCT04194775	Nofazinlimab with Lenvatinib	Lenvatinib	1st	III
NCT04344158	AK105 with Anlotinib	Sorafenib	1st	III
NCT04401800	Tislelizumab with Lenvatinib	n/a	1st	II
NCT04720716	IBI310 with Sintilimab	Sorafenib	1st	III
NCT04770896	Atezolizumab with Lenvatinib or Sorafenib	Lenvatinib or Sorafenib	2nd	III
NCT03439891	Sorafenib with Nivolumab	n/a	1st	II
NCT03211416	Sorafenib with Pembrolizumab	n/a	1st or 2nd	I/II
NCT04039607	Nivolumab with Ipilimumab	Lenvatinib or Sorafenib	1st	III

**Table 3 ijms-25-01259-t003:** Ongoing CAR T-cell trials [72,73,74,75,76,77,78,79,80,81,82,83].

Trial ID	CAR T-Cell Target	Phase
NCT05003895	GPC3	I
NCT04864054	GPC3	I/II
NCT05926726	GPC3	I
NCT05620706	GPC3	I
NCT05783570	GPC3	I
NCT05120271	GPC3	I/II
NCT03198546	GPC3	I
NCT04951141	GPC3	I
NCT05131763	NKG2DL	I
NCT05028933	EPCAM	I
NCT04502082	AFP	1/II
NCT03132792	AFP	I

**Table 4 ijms-25-01259-t004:** Trials involving NK cells for HCC [88,89,90,91,92,93].

Trial ID	Agent	Phase
NCT05845502	CAR-NK	I
NCT02839954	CAR-NK	I/II
NCT04162158	NK cells	I/II
NCT05171309	Camrelizumab, Apatinib, and NK cells	II
NCT02562963	NK cells	I/II
NCT05040438	NK cells and hepatic artery infusion pump	II

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
