# Peer review of "Recent Progress in Systemic Therapy for Advanced Hepatocellular Carcinoma"

_ijms, 2024, doi:10.3390/ijms25021259_

Round 1

Reviewer 1 Report

Comments and Suggestions for Authors

Dear author,

Thank you for allowing me to review your manuscript. I provide some comments to improve the readability and understanding of readers. 1. Describe the major hurdles of treating hepatocellular carcinoma (HCC) with a novel understanding of new treatment options in the introduction section with proper schematic diagrams. 2.

2. In the section "Evolution of First-Line Treatment" with schematic diagrams.

3. Please add a new section on drug resistance in hepatocellular carcinoma (HCC) with the advantages and limitations of drugs.

4. Please add more schematics for understanding your manuscript.

Thank you so much.

Best Regards

Thank you for allowing me to review your manuscript. I provide some comments to improve the readability and understanding of readers. 

  1. Hepatocellular carcinoma (HCC), a primary liver cancer, poses a formidable challenge in the realm of oncology due to its high incidence and resistance to conventional therapies. Please describe how searching for novel treatment options has become imperative to improve patient outcomes. In this comprehensive review, please delve into the major hurdles associated with treating HCC and explore the evolving landscape of first-line treatments; Shaping light on the promising developments with accompanying schematic diagrams will provide a visual aid to facilitate a better understanding of the intricate processes involved.

2. Major Hurdles in HCC Treatment: Hepatocellular Carcinoma (HCC), a primary liver cancer, poses a formidable challenge in the realm of oncology due to its high incidence and resistance to conventional therapies. The search for novel treatment options has become imperative to improve patient outcomes. In this comprehensive review, we delve into the major hurdles associated with treating HCC and explore the evolving landscape of first-line treatments, shedding light on the promising developments. Please describe schematic diagrams that will provide a visual aid to facilitate a better understanding of the intricate processes involved.

3. Please describe the Evolution of First-Line Treatment in the following options:

The landscape of first-line treatments for HCC has undergone a paradigm shift in recent years, with advancements in targeted therapies and immunotherapies. Schematic diagrams will chart the evolution of these treatments, providing a visual narrative of the key milestones.

Targeted Therapies:
Targeted therapies, such as tyrosine kinase inhibitors and immune checkpoint inhibitors, have emerged as promising avenues for HCC treatment. Schematic representations will elucidate the molecular mechanisms behind these therapies, highlighting their specificity in targeting cancer cells while sparing healthy tissues. Insights into the signaling pathways involved will underscore the rationale behind these therapeutic approaches.

Immunotherapies:
The advent of immunotherapies, particularly immune checkpoint inhibitors, has introduced a new dimension in HCC treatment. Schematic diagrams will illustrate how these agents unleash the immune system's potential to recognize and eliminate cancer cells. Key checkpoints and their modulation will be visually depicted to underscore the intricacies of immunotherapeutic interventions in HCC.

3. Please describe the Drug Resistance in HCC the following options:

Despite the progress made in first-line treatments, drug resistance remains a formidable challenge in HCC management. A new section will explore the nuances of drug resistance, unraveling the advantages and limitations associated with currently available drugs.

Benefits:
Understanding the advantages of existing drugs in HCC is crucial for tailoring treatment strategies. Schematic representations will elucidate the mechanisms through which drugs exert their anti-cancer effects, emphasizing their role in suppressing tumor growth and angiogenesis. Highlighting the positive aspects of these drugs will provide a comprehensive perspective on their therapeutic potential.

Limitations:
Conversely, the section will delve into the limitations of current drugs, including the development of resistance mechanisms and adverse effects. Schematic diagrams will elucidate the molecular pathways leading to drug resistance, emphasizing the need for innovative approaches to overcome these challenges. Insights into the limitations will pave the way for developing more effective and durable therapeutic strategies.  

4. In conclusion, please provide a comprehensive overview that provides a detailed exploration of the major hurdles in HCC treatment, the evolving landscape of first-line therapies, and the intricate dynamics of drug resistance. Schematic diagrams serve as valuable tools to enhance understanding, making this review an invaluable resource for clinicians, researchers, and stakeholders invested in advancing the field of HCC therapeutics.

Reviewer 2 Report

Comments and Suggestions for Authors

Sadagopan et al. reviewed recent progress in systemic therapy for advanced hepatocellular carcinoma.

The manuscript is interesting and organized well. I have some minor concerns that need to be addressed before publication.

1.      Using clinical data, authors should explain the side effects of existing therapy in detail.

2.      With recent literature updates, the introduction should be revised by adding molecular mechanisms and factors responsible for hepatocellular carcinoma.

3.      Hormonal and immunotherapy should discussed in detail using recent discoveries in this field, inducing their advantages and disadvantages.

4.      The conclusion should be revised to focus on the future direction for the therapy of hepatocellular carcinoma.

Comments on the Quality of English Language

Minor editing of English language required
